# Accuracy in Cement Hydration Investigations: Combined X-ray Microtomography and Powder Diffraction Analyses

**DOI:** 10.3390/ma14226953

**Published:** 2021-11-17

**Authors:** Inés R. Salcedo, Ana Cuesta, Shiva Shirani, Laura León-Reina, Miguel A. G. Aranda

**Affiliations:** 1Servicios Centrales de Apoyo a la Investigación, Universidad de Málaga, 29071 Málaga, Spain; inesrs@uma.es (I.R.S.); lauralr@uma.es (L.L.-R.); 2Departamento de Química Inorgánica, Cristalografía y Mineralogía, Universidad de Málaga, 29071 Málaga, Spain; a_cuesta@uma.es (A.C.); shiva_shirani@uma.es (S.S.)

**Keywords:** Portland cements, Rietveld method, 3D image segmentation, microstructure, porosity

## Abstract

Cement hydration is a very complex set of processes. The evolution of the crystalline phases during hydration can be accurately followed by X-ray powder diffraction data evaluated by the Rietveld method. However, accurate measurements of some microstructural features, including porosity and amorphous content developments, are more challenging. Here, we combine laboratory X-ray powder diffraction and computed microtomography (μCT) to better understand the results of the μCT analyses. Two pastes with different water–cement ratios, 0.45 and 0.65, filled within capillaries of two sizes, ϕ = 0.5 and 1.0 mm, were analysed at 50 days of hydration. It was shown that within the spatial resolution of the measured μCTs, ~2 μm, the water capillary porosity was segmented within the hydrated component fraction. The unhydrated part could be accurately quantified within 2 vol% error. This work is a first step to accurately determining selected hydration features like the hydration degree of amorphous phases of supplementary cementitious materials within cement blends.

## 1. Introduction

Cements are multi-phase materials and they can be understood well when studied by a multi-technique approach [1], mainly: (i) X-ray fluorescence for elemental analysis; (ii) X-ray powder diffraction, using the Rietveld method, for quantitative phase analysis; and (iii) a set of techniques for textural characterization (mainly Blaine air permeability, laser diffraction and nitrogen sorption isotherm). The understanding of cement hydration features is less developed. Hydration can take place in pastes, mortars or concretes. Here, the discussion is restricted to the hydration of cement pastes, as mortars and concretes have an additional layer of complexity due to the aggregates.

The hydration of cement pastes is itself very complex due to the interplay of several parameters. Firstly, the starting materials are multiphase with a wide range of particle sizes. Secondly, there is an increasing trend of using supplementary cementitious materials (SCMs) [2] and organic and inorganic admixtures [3]. Thirdly, the hydration conditions can vary: water-to-cement (w/c) mass ratio, temperature and pressure. And fourthly, after some time, hydration conditions can change and important processes for durability could take place such as carbonation or chloride ingress [4].

It is acknowledged that many characterization techniques are necessary to fully understand the features of cement hydration. Interested readers are referred to an excellent book where many are thoroughly treated (i.e., calorimetry, shrinkage measurements, thermal analysis, nuclear magnetic resonance (NMR), mercury intrusion porosimetry, electron microscopies, etc.) [1]. Our work is framed within a current trend of developing procedures that are reliable, relevant and rapid [5]. Powder diffraction and μCT do not require sample preparation once the capillaries are filled. They also have the potential for automation of the three steps in the data management pipeline: collection, processing and analysis. Finally, other techniques give very valuable information about cement hydration, and here, solid-state nuclear magnetic resonance spectroscopy stands out [6]. Magic angle spinning techniques, with different approaches, give local structure information and the evolution of different phases mainly through spectral deconvolution. For instance, the hydration degrees from ^29^Si MAS NMR were used to model the hydration kinetics for Portland cement–metakaolin–limestone blends [7]. Moreover, MAS NMR, in combination with other techniques, is widely used to follow the hydration of complex cement blends, see for instance [8,9]. Additionally, ^1^H NMR relaxometry is being used to quantify the different types of water present in pastes and their evolution with hydration [10,11].

After more than 20 years of research, the quantitative phase analysis of crystalline phases in hydrating cements by X-ray powder diffraction is well established [12,13,14,15,16]. Furthermore, the overall amorphous content can also be determined by using internal [17] or external [18] standard approaches. Conversely, and after more than 10 years of research, the quantitative analysis of the hydrating components and microstructures by X-ray computed microtomography (μCT) [19] is still challenging [20,21,22,23,24]. Interested readers are referred to the previous reviews to learn more about the different cementitious features that can be studied/followed and eventually quantified by μCT. Next, we will focus on the accuracy of the results obtained by the two approaches.

Accuracy in Rietveld quantitative phase analysis (RQPA) of crystalline phases in (dry) cements have been studied by preparing artificial samples [25], by a round Robin approach [15,26] and also by analysing a NIST cement reference material [27]. For hydrating cements, accuracy was initially studied by comparing the RQPA results with independent techniques including thermal analysis for quantifying the portlandite contents [28]. The main advantages of the Rietveld method applied to the analysis of cement hydration are: (i) the evolution of every crystalline phase can be independently followed if its crystal structure is known [13], (ii) data analysis can be automatized with scripts and/or appropriate software [29], (iii) it does not require calibration curves, and iv) it profits from the whole powder pattern which minimizes the consequences of systematic errors like preferred orientation. The main drawbacks are: (i) without using an internal standard, the results are normalized to 100% of crystalline phases and hence all amorphous content information is missed; (ii) using the internal standard method [17], only an overall phase fraction is determined which is not sufficient in cement hydration where usually at least three amorphous components are present (free water, C-S-H gel and amorphous components in the SCMs, like metakaolin in calcined clays); and (iii) microabsorption could lead to severe errors [30,31]. Furthermore, C-S-H quantification is key for investigating cement hydration but it cannot be accurately obtained by RQPA. Alternative methods like PONKS (partial or no known crystal structure) has to be employed [32,33].

To the best of the authors’ knowledge, there are very few works devoted to determine/estimate the accuracy of μCT applied to cementitious materials. Here, an initial work should be noted in which standard mixtures were prepared from four single phases: C-S-H gel, calcium hydroxide, monosulfate (AFm) and ettringite (AFt) [34]. The mixtures were studied by synchrotron (monochromatic) μCT and the segmentations were obtained by global thresholding of the reconstructed tomograms. Most binary mixtures were segmented with accuracies close to 20% as the attenuations were quite similar. Moreover, the uncertainties in quaternary mixtures were much higher, in some cases above 100%, relative error, when compared to the expected vol% [34]. The same authors reduced the relative errors to ~50% by employing two local segmentation algorithms: Markov random fields and watershed [35]. However, this approach, employing standard mixtures from mixing pure phases, does not take into account the fine intermixing during precipitation which results in larger systematic errors, because of partial volume effects.

Finally, and to the best of our knowledge, X-ray powder diffraction and computed microtomography have not been combined for quantifying cementitious binders. We address this knowledge gap in order to estimate the accuracy in the results of the μCT analyses by determining the amounts of three sets of components based on their attenuations: (i) porosity, (ii) hydration products, and (iii) unhydrated cement phases. Here, we have analysed a hydrating cement with two water-to-cement mass ratios (w/c) within capillaries of two diameters at about 50 days of hydration by two X-ray techniques applied to the same regions of the capillaries. It is shown that anhydrous cement phases and hydrated products can be accurately analysed by μCT. It is also demonstrated that the free water is segmented, when using the global thresholding approach, within the hydrated components.

## 2. Materials and Methods

### 2.1. Portland Cement

The commercial Portland cement used was a CEM I 42.5 N type which conforms to EN 197–1, from Financiera y Minera, (Heidelberg Cement Group, Malaga, Spain). Full characterization of this cement has been reported elsewhere [36], hence, only a summary is given here. The elemental analysis was 62.89, 19.75, 4.98, 3.41, 3.37, 1.53, 1.05 and 0.30 wt% for CaO, SiO_2_, Al_2_O_3_, Fe_2_O_3_, SO_3_, MgO, K_2_O and Na_2_O, respectively. The mineralogical analysis is given in Table 1. The air permeability, measured by the Blaine assay, was 3700 cm^2^/g.

### 2.2. Paste Preparation

Cement pastes were prepared with deionized water (w/c = 0.45 and 0.65) by mechanical stirring at a speed of 800 rpm for 90 s. After waiting for 30 s, another identical stirring step was undertaken. The pastes were injected into glass capillaries of diameters of 0.5 and 1.0 mm with a syringe and sealed with UV hardening resin. In addition, and to prepare reference samples that can also be characterised by thermal analysis, the same pastes were poured into polytetrafluoroethylene (PTFE) cylinder moulds, 35 mm long and 10 mm in diameter, that were hermetically closed and rotated at 16 rpm, at 20 ± 1 °C. The samples were kept in the moulds for 24 or 48 h for the pastes with w/c = 0.45 and 0.65, respectively. Then, the cylinders were demoulded and placed into a bottle of water at 20 ± 1 °C to be used at the selected hydration age. The bottle was tightly closed to minimise carbonation and the amount of water was the minimum to minimise calcium leaching. For a fraction of the cylinders, the hydration of the samples was stopped before their characterisation by manually grinding and washing twice with isopropanol and finally with diethyl ether. The two hydration-arrested powder samples were gently ground with the internal standard, 20 wt% of quartz, for RQPA.

### 2.3. Thermogravimetric Analysis

A SDT-Q600 analyser from TA instruments (New Castle, DE, USA) was used to collect the thermogravimetric (TGA) traces for the four samples, i.e., two hydration-arrested and two neat pastes. The temperature was varied (10 °C/min) from RT to 40 °C, held at 40 °C for 30 min and then raised up to 1000 °C at the same rate. Measurements were made in open platinum crucibles under synthetic air flow. The weight loss from 45 to 550 °C was assigned to chemically bounded water [37,38] and the loss from 550 to 1000 °C was considered as CO_2_. The free water (FW, wt%) was calculated from the TGA traces of the hydration-arrested pastes by assuming that the chemically combined water was responsible for the weight loss from 45 to 550 °C. The FW was obtained by subtracting this value from the total added water (nominal value).

### 2.4. Laboratory X-ray Powder Diffraction (LXRPD) and Data Analysis

The pastes within the capillaries were measured on a D8 ADVANCE (Bruker AXS) diffractometer, located at SCAI at UMA (Malaga, Spain), using monochromatic Mo-Kα_1_ radiation (λ = 0.7093 Å). The optics configuration was a primary monochromator, a focusing mirror, 1 or 2 mm anti-scatter slits (for 0.5 and 1 mm of capillary diameters, respectively), and 2.5° Soller slits for the incident and transmitted beams. The X-ray detector was an EIGER (from DECTRIS, Baden, Switzerland) specially designed and optimized for Mo anodes, which was used with an aperture of 7 × 21 degrees, working in VDO mode. Data were collected from 3 to 35° (2θ) during approximately 4 h. The tube worked at 50 kV and 50 mA. The pastes from the cylinders were measured in transmission geometry placing the powders between two Kapton foils. Measurements were carried out in the same diffractometer with the same conditions and very similar optics (2 mm incident anti-scatter slit).

All the LXRPD patterns were analysed by the Rietveld method using a GSAS software package [39] by using a pseudo-Voigt peak shape function with the asymmetry correction included [40,41]. The refined overall parameters were: background coefficients, zero-shift error, phase scale factors, unit cell parameters, peak shape parameters and preferred orientation coefficient if needed. The employed crystal structures are given elsewhere [42,43]. The determination of amorphous and crystalline non-quantified (ACn) content was performed by the internal standard methodology [17] mixing the powders with ~20 wt% of quartz.

### 2.5. Laboratory X-ray Computed Microtomography (μCT) and Data Processing and Analysis

The μCT images were acquired on a SKYSCAN 2214 (Bruker) scanner at SCAI at UMA. Images were obtained with an X-ray tube using a LaB_6_ source (operated at 55 kV and 130 μA) and employing a 0.25 mm Al foil to reduce beam hardening. The capillaries were rotated 360° during data acquisition. Images were taken every 0.2°, i.e., 1801 frames, with an exposure time of 2.0 s resulting in an overall recording time of 4.72 h per CT. The geometrical settings were a source to object distance of 14.480 mm and an object to detector distance of 315.449 mm. The detector system was the CCD3 which has a physical pixel size of 17.4 μm (binning 1 × 1), which together with the geometric settings resulted in a voxel size of 0.80 μm.

NRecon (version 2.1.0.1, Bruker) was used for the reconstruction of the CTs. Dataviewer (Bruker) software was used for 3D visualization. The analysed volumes were cylinders with a height of ~1215 μm, and diameters of ~450 and 900 μm, for ϕ = 0.5 and 1.0 mm glass capillaries, respectively. Many filtering methods have been developed to reduce the background noise of the images [44,45]. Here, a 3 × 3 × 3 median 3D filter was applied to the four μCTs with Fiji program [46]. This was to reduce image noise and to even out contrast variations between adjacent slices. This procedure slightly sharpens the greyscale value histograms. Fiji was also used for data analysis. Image segmentation was performed by applying a global threshold approach with boundaries determined as described in the results section. Rendered volumes displaying the outcomes of the segmentations were pictured using CTVox software (Bruker).

## 3. Results

### 3.1. Thermal Analysis

Thermal analysis data were collected for the powder samples cast as cylinders with w/c = 0.45 and 0.65, i.e., with and without arresting the hydration (four traces). Figure 1. shows the TGA profiles for the neat pastes, i.e., without arresting the hydration, which allows extracting three relevant values: (i) total weight loss which corresponds to all the water plus the contributions from limestone and any hydrated phase within the pristine cement; (ii) the portlandite –CH or Ca(OH)_2_–; and (iii) calcite–C$\bar{C}\;$or CaCO_3_– contents, which are referenced to the neat pastes. This allows a direct comparison with the RQPA results, see below. The percentages are calculated in a two-step process, (a) the weight loss is obtained by the tangential method [47], and (b) the value is converted to weight percentage using the 4.11 or 2.27 factors, for portlandite or calcite, respectively. The FW percentages, given in Table 1 and Table 2, were determined from the TGA curves of the hydration-arrested pastes as detailed in the experimental section.

### 3.2. Laboratory X-ray Powder Diffraction Study

For a given paste, three LXRPD patterns were collected: two for the pastes within the capillaries (ϕ = 0.5 and 1.0 mm) and a third one for the paste casted as cylinder. This last sample underwent hydration stoppage and it was gently milled with the internal standard. The Rietveld fits for the three patterns, w/c = 0.45 pastes, are displayed in Figure 2 and the results are reported in Table 1.

For reporting the data shown in Table 1, some additional calculations/assumptions were made. Firstly, this table contains the FW determined from the TGA as detailed above. Secondly, and as an internal standard was used, an overall ‘amorphous’ amount was determined. The ACn values reported in Table 1 are the subtraction of the FW to the ACn contents. The value for ACn is mainly, but not only, C-S-H gel. Thirdly, as the capillaries cannot contain an internal standard (to avoid the filler effect and to carry out μCT of the pastes), here it is assumed that they contain the same fractions of FW and ACn as the analogous cylinder samples. Furthermore, the external standard method cannot be used for capillary samples as it is not possible to control the packing factor.

The results for the pastes hydrated with w/c = 0.65 at 20 °C for 50 days are given in Table 2 and the corresponding Rietveld plots are displayed in Figure 3. The same calculations/assumptions, described above for the w/c = 0.45 pastes, were followed here. As the RQPA results for the capillaries can be referenced to 100 g of paste, the overall hydration degree of the pastes can be calculated. From data in Table 1 and Table 2, the overall hydration degrees of the ϕ = 1.0 mm pastes for w/c = 0.45 and w/c = 0.65 are 78.3 and 82.3 %, respectively.

### 3.3. Laboratory X-ray Computed Microtomographic Study

Figure 4 displays representative tomographic orthoslices showing glass capillary cross sections for the four studied pastes. In all reconstructions, the unhydrated cement products (UHP) are the brightest set of phases as they are the most absorbing components. The porous regions (air and water) are the darkest phase, as their attenuations are the lowest. Hydrated products (HP), which included limestone because of its attenuation, have intermediate absorption and hence, middle grey-scale values. A first qualitative analysis indicates that the pastes with w/c = 0.65, see Figure 4b,d have more porosity, as expected, than those with w/c = 0.45, Figure 4a,c. The determined free water contents were 25 and 14 wt%, respectively. Shrinkages are evident in the μCTs as there is air between the glass capillary walls and the binders. However, the study of this shrinkage and its evolution with hydration age was out of the scope of the present work.

To illustrate the spatial resolution of the reconstructed tomographic data, Figure 5 displays two enlarged views. The plot profiles, shown in Figure 5b,d, allow us to visualize the three main components: (i) porosity (water and air) with grey-scale values ranging ~0–55; (ii) hydrated components, ~55–100/120; and (iii) unhydrated cement particles, >100/120. It can also be seen that relatively small pores, size ~4 μm, are readily measured which highlights the good spatial resolution of the acquired data. From several measurements, the spatial resolution is estimated as ~2 μm. It is noted that although the voxel size was 0.80 μm, the spatial resolution can be/is poorer.

A first data analysis can be carried out by inspecting the greyscale histograms (GSHs). The linear attenuation coefficient μ is mapped in the reconstructed tomograms, where the higher is the μ value, the whiter is the corresponding greyscale value. Figure 6 displays the GSHs for the four studied capillaries. For three out of the four capillaries, there are clear valleys between the hydration products and the unhydrated cement products. For selecting the greyscale thresholds in the absence of valleys between the contributions, the tangent-slope approach was used [48]. The grayscale threshold separating two components was defined as the point where a change of tangent-slope for both contributions occurred. The procedure, and the derived values, are given in Figure 6.

It is evident in Figure 6 that for a given capillary size, the increase in added water displaces the maximum of the GSH for the HP to less absorbing regions. This observation plus the absence of a large contribution of water in the pore regions, greyscale values lower than ~60, indicates that most free water contribution are within the hydration product peak.

The four recorded μCTs were segmented by global thresholding using Fiji software. The thresholds for the component assignment were derived as detailed above and it is acknowledged that the manual threshold choice may slightly affect the accuracy of the results of the segmentation. The errors related to this approach have been previously estimated as ~2 vol% [49]. The segmentation results are listed in Table 3. Figure 7 displays the rendered volume of the segmented tomograms. As the porosity contribution was mainly air, Table 3 also reports the segmented values but renormalized just taking into account the hydrated products and the unhydrated cement fraction labelled as rHP and rUHP, respectively. As expected, the overall degree of hydration is larger for the pastes hydrated with larger w/c ratio, i.e., w/c = 0.65. Furthermore, for the same paste (i.e., the same w/c ratio) the results from the two capillary sizes are the same within the variability of the measurements, the differences were lower than 1.5 vol%. Porosity, see Figure 7, is mainly observed close to the glass walls due to chemical shrinkage. The segmented porosity within the interior of the capillaries, because of its tiny dimensions, can only be observed in the w/c = 0.65 pastes.

## 4. Discussion

In this work, accuracy in cement hydration is studied in three stages. Firstly, thermal analysis results were compared to those obtained from RQPA of the pastes casted within cylinders. Secondly, the RQPA results from the three powder samples hydrated with the same w/c ratio were compared. Finally, the results from the analyses of the μCT were compared to those obtained from the capillaries by LXRPD. All laboratory X-ray data sets (diffraction and microtomography) were taken in approximately the same regions of the capillaries. Note that this study was carried out at 50 days of hydration. This choice was larger than the standard maturing time, i.e., 28 days, for two reasons: (a) to ensure that longer hydration times do not lead to significant carbonation; and (b) that low unhydrated cement product contents could be measured to an acceptable accuracy level.

### 4.1. Comparison Involving the Thermal Analyses

TA was chosen because it is a technique readily available, widely used for analysing cement hydration, which allows measuring at least three relevant parameters: (i) portlandite content, (ii) chemically bound water, and (iii) calcium carbonate content [47]. It had already been used in 2004 to verify the findings from Rietveld quantitative phase analysis of pastes by comparing the portlandite contents determined from the two methods and also by backscattered electron images [28]. TA, in combination with RQPA [50], has been used for characterizing the hydration of many types of cements, see for instance: (i) limestone-Portland blends [51]; (ii) limestone-fly ash-Portland ternary blends [37]; and (iii) limestone-fly ash-blast furnace slag-Portland quaternary blends [52]. Furthermore, the amount of measured portlandite depends upon several factors including the mineralogy of the employed cement, the hydration temperature and the use of additions. For comparisons, the content should be referenced to the same basis, for instance 100 g of paste. In this context, this cement was employed in an independent study by the authors and the portlandite contents determined at 28 days for a paste hydrated at 20 °C with w/c = 0.50 by TA and RQPA were 12.2 and 13.0 wt%, respectively [49]. These data compare very well with the portlandite percentages given in Table 1 and Table 2 and Figure 1 (ranging 10.7–12.4 wt%), taking into account that a small consumption of portlandite could take place at longer hydration ages to give AFm-type phases.

#### 4.1.1. Thermal Analysis and Nominal Compositions

On the one hand and for the non-arrested samples, the total weight loss expected for the pastes corresponds to the added water plus the losses from the decomposition of calcite and bassanite. For the w/c = 0.45 paste, the measured value was 29.5%, see Figure 1, which agrees relatively well with the expected overall weight loss, 32.5%. For the w/c = 0.65 paste, the measured value, 39.2%, also agrees well with the expected one, 41.0%. The gentle grinding before the thermal analysis could be the responsible for the small difference, ~2 wt%, which is very likely due to water loss at that stage. On the other hand, the TGA determined C$\bar{C}\;$content, ~5 wt% for the w/c = 0.65 paste, is larger than the expected value, ~2.2 wt%. This minor carbonation is not negligible but it does not importantly influence the comparisons carried out next.

#### 4.1.2. Thermal Analysis and Rietveld Quantitative Phase Analysis (RQPA) of the Pastes from the Cylinders

The portlandite contents allow comparing the results from these two analytical approaches. For the w/c = 0.45 paste, the CH content from TGA was 10.7 wt%, see Figure 1. The corresponding value for the paste within the cylinder, from RQPA, was 10.8 wt%. Furthermore, the agreement for the w/c = 0.65 pastes is also good, 10.9 and 10.0 from TGA and RQPA, respectively. The Rietlved value is slightly lower very likely due to the small carbonation of this sample. In fact, the C$\bar{C}\;$content from RQPA was 5.5 wt%, which is higher than the value coming from the pristine cement, ~2.2 wt% after dilution.

The successful comparison between the values from TGA and RQPA for the cylinder-casted pastes indicates that the RQPA for these samples were accurate and they can be used as reference “ground truth” for the next comparison: RQPA results from pastes hydrated within tiny capillaries and the reference samples.

### 4.2. Comparison of RQPA Results Obtained in Different Conditions for the Same Pastes

#### 4.2.1. Considerations about the Width of the Capillaries for Cement Paste Analysis

It must be noted that the final goal is to combine data taken in the same regions of the same capillaries by LXRPD and μCT. The optimal size of the capillary is a trade-off between two opposite effects. On the one hand, the tiniest capillary results in lower overlapping in the powder patterns due to the narrower diffraction peak widths and better resolution in the μCT as the field of view is smaller. However, very narrow capillaries result in poor particle statistics and also it is more difficult to ensure the correct (nominal) w/c ratio [53,54] in the probed volume as microbleeding could develop. On the other hand, wider capillaries have better particle statistics theoretically enabling better accuracy, but if the capillaries are very wide, the LXRPD patterns as well as the μCTs will have poorer resolution, which could lead to larger errors. Because of these opposing effects, capillaries of two different sizes, ϕ = 0.5 and 1.0 mm were filled with exactly the same pastes.

#### 4.2.2. Comparison of the RQPA Results or the Same Pastes in Different Capillary Sizes

Firstly, the results for the three patterns containing the w/c = 0.45 pastes will be discussed. The data reported in Table 1 show a very good agreement between the cylinder and the ϕ = 1.0 mm capillary analyses. For instance, the C_3_S contents are 4.3 and 4.2, for the cylinder and ϕ = 1.0 mm samples, respectively. This agreement is also evident in the formed phases, i.e., the CH contents are 10.8 and 11.6 wt%, respectively. Unfortunately, the agreement with the ϕ = 0.5 mm capillary results is not that good. The C_3_S and CH contents are 7.5 and 8.3 wt%, respectively. It seems clear that the degree of hydration for the unhydrated cement phases is lower for the w/c = 0.45 paste within the narrow capillary. We justify this because ensuring the nominal w/c ratio within tiny capillaries is very challenging, as previously reported [53,54]. Finally, it is worth noting that ettringite contents for the sample within the ϕ = 1.0 mm capillary and the hydration arrested paste were 11.2 and 10.8 wt%, respectively. This means that the hydration arresting step has not significantly decreased the crystalline ettringite content.

Secondly, the results for the three w/c = 0.65 patterns will be considered. Table 2 shows a very good agreement for the contents derived for the three pastes. For instance, the ettringite fractions were 10.2, 11.1 and 9.8 wt% for the pastes ϕ = 0.5 mm, ϕ = 1.0 mm and cylinder, respectively. In this case, with a relatively large water/cement ratio, the RQPA results for the sample within the ϕ = 0.5 mm is accurate as microbleeding/segregation seems less important.

It is noted that carbonation has not taken place in any of the two wide (ϕ = 1.0 mm) studied capillaries, see Table 1 and Table 2. Altogether, and because low water/cement ratios are very common in field applications, it is concluded here that relatively thick capillaries are needed to have RQPA results as accurate as possible.

### 4.3. Comparison of the RQPA and μCT Results for the Pastes within the Same Capillaries

The mass densities of the crystalline phases are known and are given in Table 1. For the amorphous phases, we have used the approximation of the density being the same than that of C-S-H gel, which is variable at the microscale, i.e., ~2.0 g/cm^3^ [55], and the latest value 2.11 g/cm^3^ was used here [56]. Under this approximation, the results of the RQPA, given in Table 1 and Table 2, can be transformed from wt% to vol%. The different percentages were grouped as unreacted cement products and hydrated products, including the free water, and they are given in Table 4. This table also reports the results from the segmentation of the μCT after renormalization in order not to take into account the porosity.

As can be seen in Table 4, the agreement between the values obtained from diffraction and tomography, for a given w/c ratio, is relatively good. Firstly, and as expected, both approaches yield a larger degree of reaction to the pastes with a larger w/c ratio, i.e., lower UHP content. Secondly, the agreement between the results for the same paste within capillaries of two sizes are also good. However, it is noted that any experimental variation within the narrow capillaries, i.e., capillaries with a nominal diameter of 0.5 mm, will result in larger deviations. Thirdly, the relative large disagreement for the w/c = 0.65 paste within a 1.0 mm capillary is likely due to the small fraction of UHP, which slightly increase the errors. Finally, and from the results shown in Table 4, it is estimated that the segmented values in the μCT are accurate within ~2 vol%.

## 5. Conclusions

The following main conclusions can be drawn from this work:A methodology for obtaining μCT and LXRPD datasets for a cement paste within a capillary is reported. The resulting data have the necessary quality to be quantitatively analysed by global thresholding segmentation and Rietveld approaches.For establishing this methodology, the combined X-ray tomographic and diffraction study was complemented with results from pastes casted in moulds. The thermal analysis results and the amorphous contents derived from RQPA, by using the internal standard methodology, gave the necessary information to ascertain whether the results were accurate, or not.It is concluded that the values obtained by segmentation of laboratory μCT data are accurate, within ~2 vol%, but only if free water is added to the hydrated product component. This result holds for the employed μCT main experimental conditions: 0.80 μm voxel size and ~2 μm of true spatial resolution.To minimise the consequences of unavoidable experimental variations, relatively thick capillaries are recommended. The diameters should be close to 1.0 mm.

## Figures and Tables

**Figure 1 materials-14-06953-f001:**
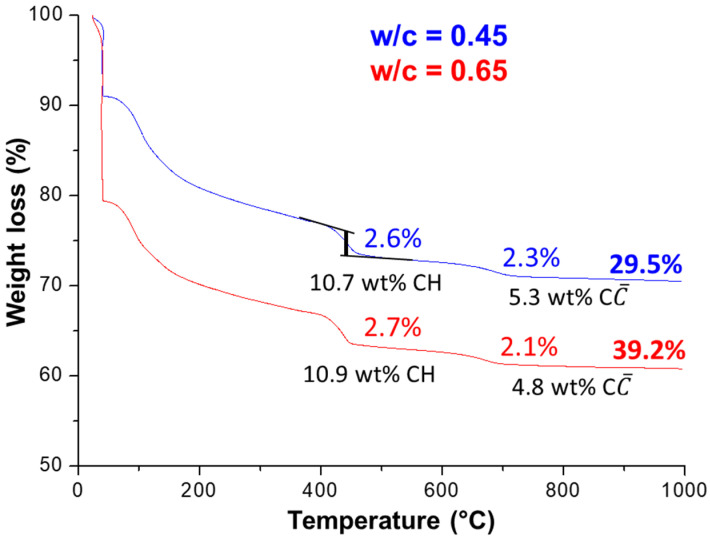
Thermal analysis data for the two neat pastes after 50 days of hydration with an isothermal heating at 40 °C for 30 min. The CH and C$\bar{C}\;$contents (wt%) are referenced to the pastes containing the free water. The total weight losses (%) are also shown.

**Figure 2 materials-14-06953-f002:**
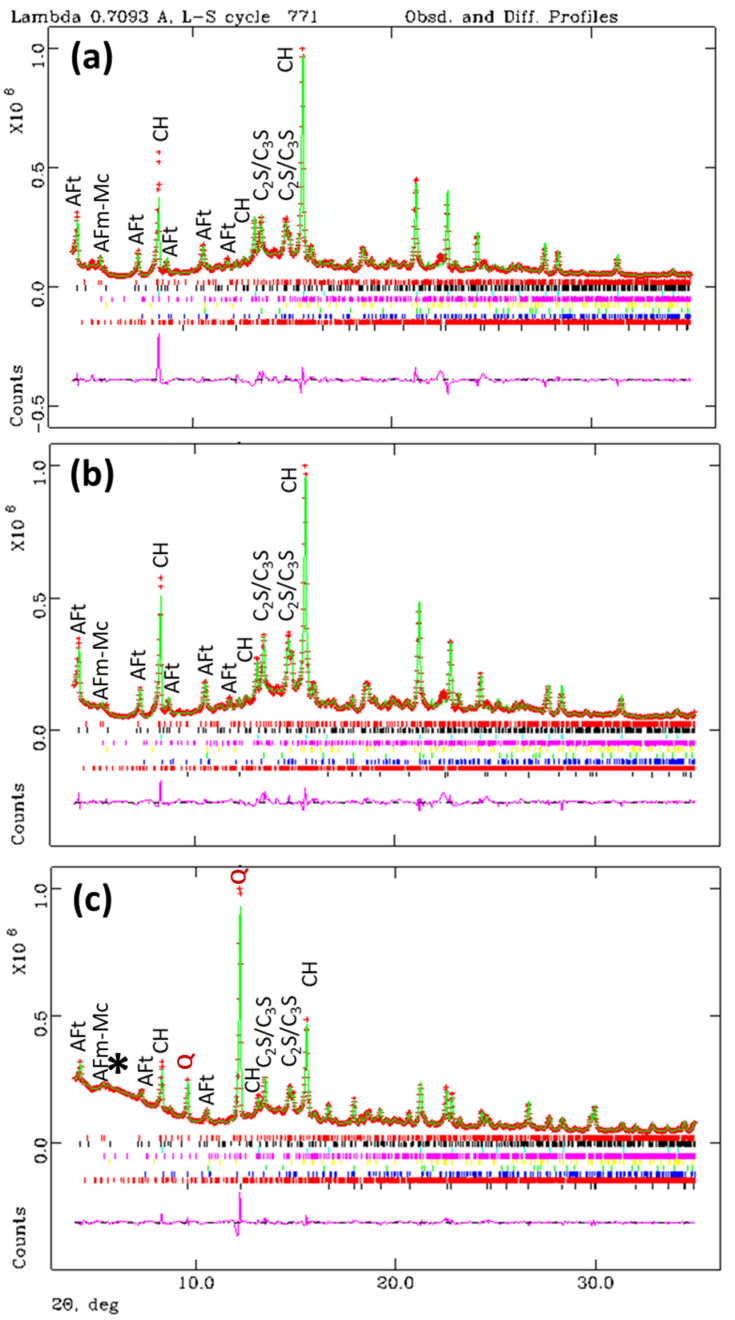
Rietveld plots, λ = 0.71 Å, for the w/c = 0.45 pastes hydrated at 20 °C for 50 days (**a**) sample within a capillary of ϕ = 0.5 mm, (**b**) sample within a capillary of ϕ = 1.0 mm, and (**c**) sample with the internal standard and between two Kapton foils. The hump due to the polymer (Kapton) is highlighted with a star. The sets of bars with different colours, above the difference curve, correspond to the crystalline phases computed. The main peaks, at low diffracting angles, are labelled. Q stands for quartz (the added internal standard).

**Figure 3 materials-14-06953-f003:**
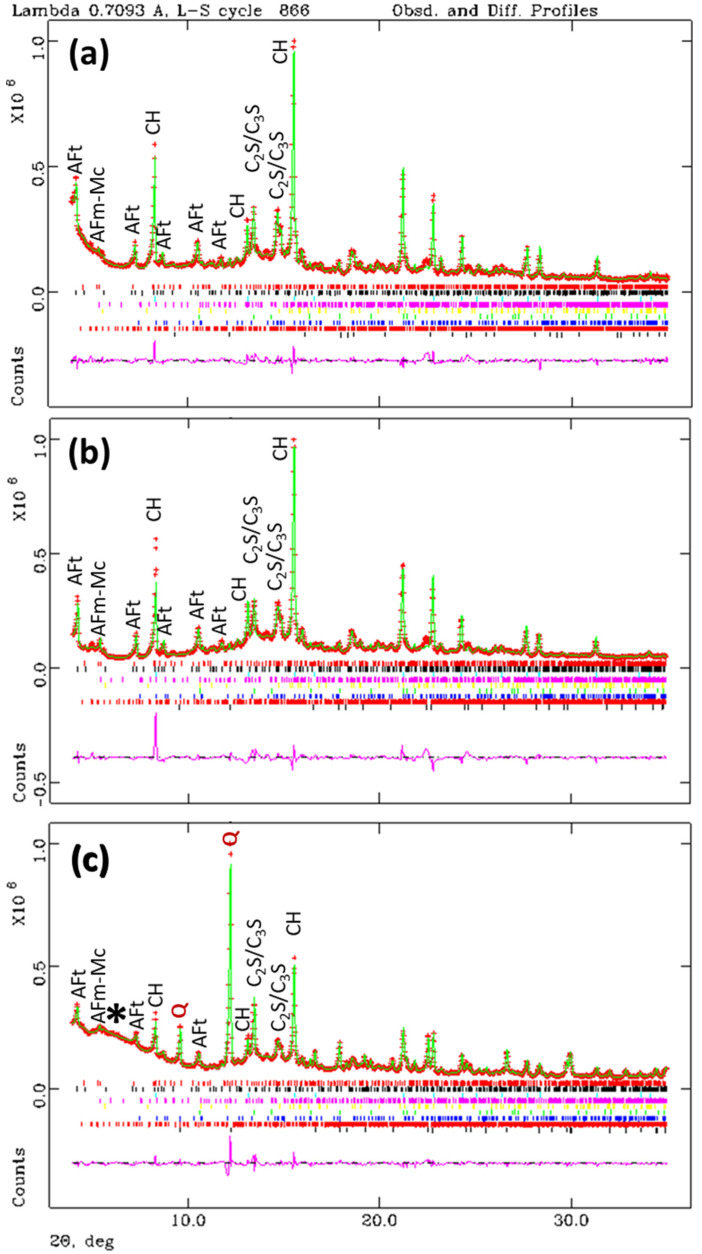
Rietveld plots, λ = 0.71 Å, for the pastes with w/c = 0.65 hydrated at 20 °C for 50 days (**a**) sample within a capillary of ϕ = 0.5 mm, (**b**) sample within a capillary of ϕ = 1.0 mm, and (**c**) sample with the internal standard and between two Kapton foils. Format and labels as in Figure 2. The hump due to the polymer (Kapton) is highlighted with a star.

**Figure 4 materials-14-06953-f004:**
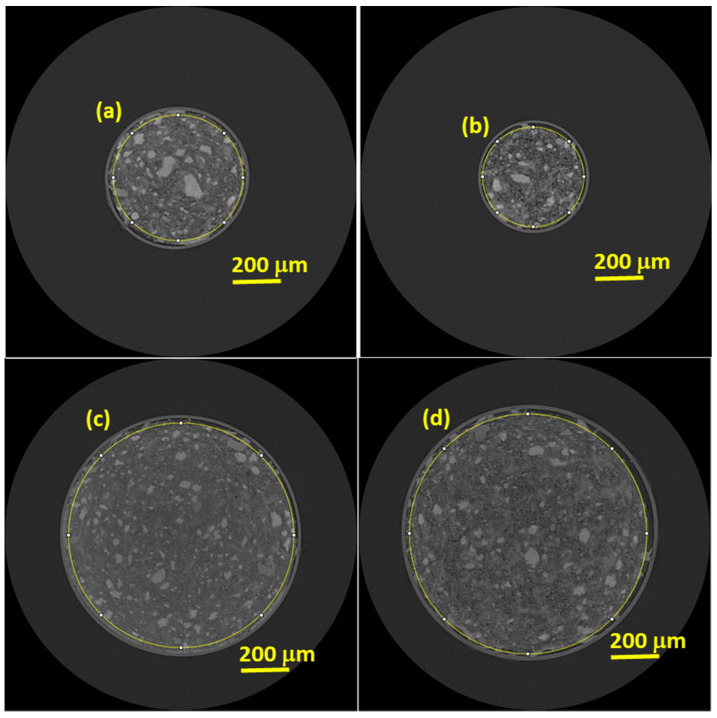
Selected reconstructed tomographic orthoslices for the studied pastes. (**a**) w/c = 0.45 paste within a capillary of nominal diameter of 0.5 mm. (**b**) w/c = 0.65 paste in a ϕ = 0.5 mm capillary. (**c**) w/c = 0.45 paste in a ϕ = 1.0 mm capillary. (**d**) w/c = 0.65 paste in a ϕ = 1.0 mm capillary. The sizes of the capillaries measured with a calibre, in the same sequence, were: 0.55, 0.45, 1.01 and 1.06 mm.

**Figure 5 materials-14-06953-f005:**
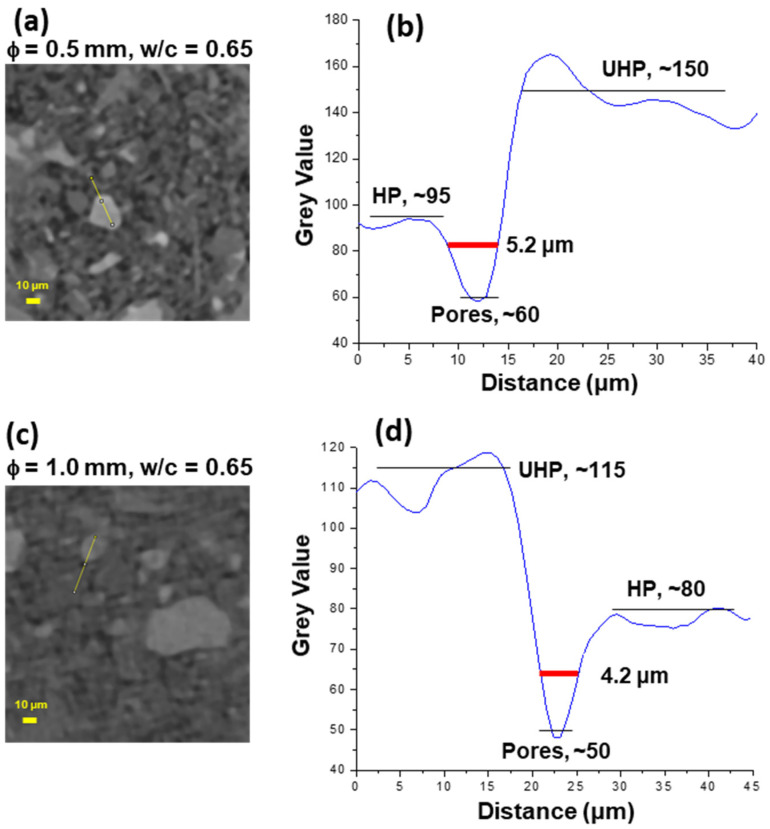
(Left) Selected enlarged views of two reconstructed slices. (Right) Plot profiles of the grey-scale values along the yellow lines depicted in the left panels which help to estimate the true spatial resolution. (**a**) The yellow line, from up to bottom, depicts hydrated products (HP), a pore and unhydrated cement products (UHP) with average grey-scale values given in (**b**). (**c**) The yellow line, from up to bottom, depicts: UHP, a pore and HP with average grey-scale values given in (**d**).

**Figure 6 materials-14-06953-f006:**
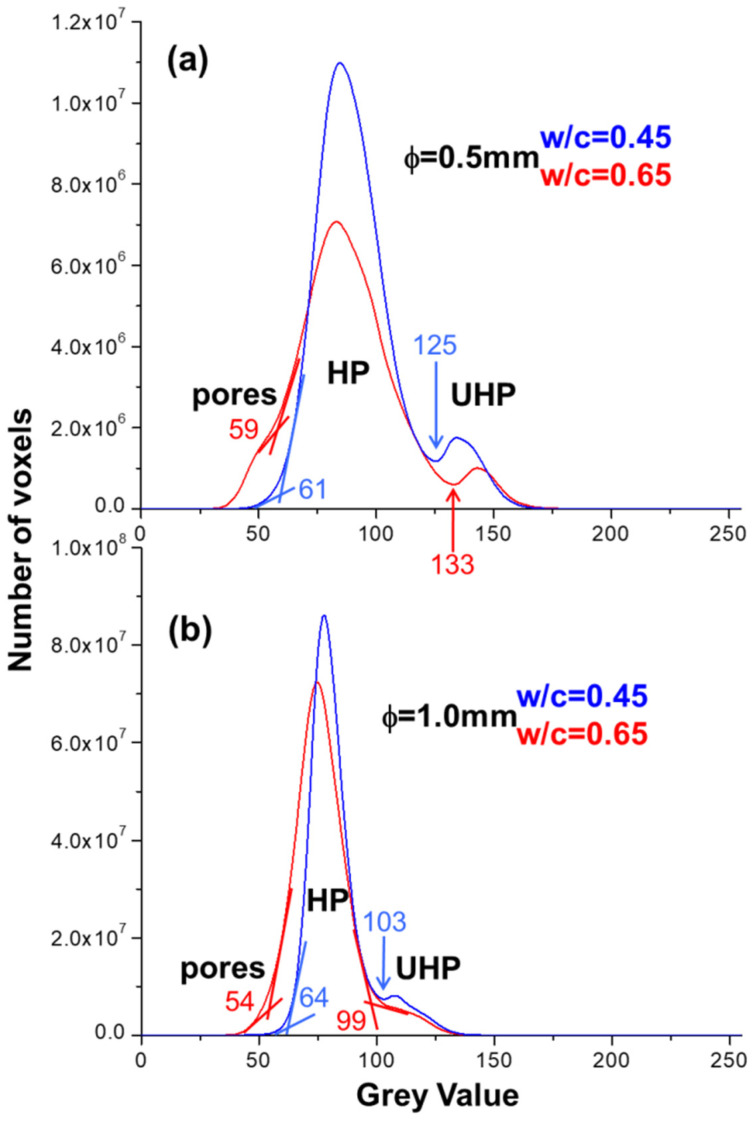
Volume of interest grey-scale histograms for (**a**) pastes within capillaries of ϕ = 0.5 mm with w/c = 0.45 and w/c = 0.65; (**b**) pastes within capillaries of ϕ = 1.0 mm with w/c = 0.45 and w/c = 0.65. The tangent-slope approach has been employed for estimating the threshold values between two components in the absence of valleys.

**Figure 7 materials-14-06953-f007:**
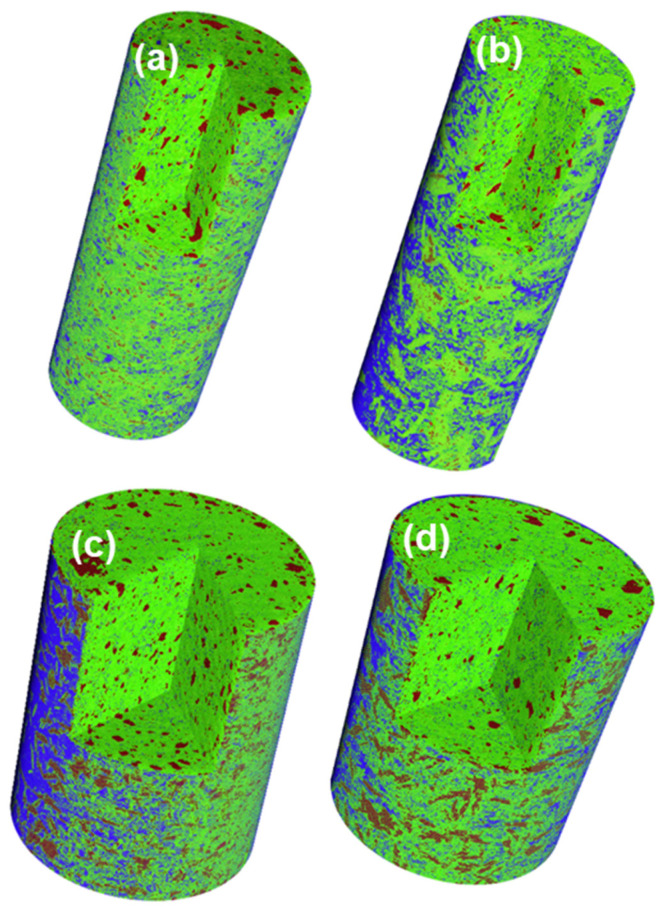
Rendered volumes displaying the results of the segmentations. Color codes: brown—unhydrated cement products, green—hydrated products, and blue—porosity (air and some remaining water). (**a**) w/c = 0.45–ϕ = 0.5 mm paste, (**b**) w/c = 0.65–ϕ = 0.5 mm paste, (**c**) w/c = 0.45–ϕ = 1.0 mm paste, (**d**) w/c = 0.65–ϕ = 1.0 mm paste.

**Table 1 materials-14-06953-t001:** Rietveld quantitative phase analysis (RQPA) results (wt%), including amorphous and crystalline non-quantified (ACn) and free water (FW) contents, for the w/c = 0.45 samples hydrated for 50 days at 20 °C and referred to 100 g of paste. Mass density of each phase is also given.

	t_0_ ^a^	t = 50 d ^b^ ϕ = 0.50 mm	t = 50 d ^b^ ϕ = 1.00 mm	t = 50 d Cylinder	*ρ* (g/cm^3^)
C_3_S	44.2	7.5	4.2	4.3	3.15
C_2_S	8.2	5.6	5.4	6.0	3.30
C_4_AF	8.3	3.4	2.4	2.5	3.73
C_3_A	3.6	1.9	0.9	1.0	3.05
$C\bar{S}$H_0.5_	0.9	--	--	--	2.71
$C\bar{C}$	2.8	4.7	2.1	2.8	2.71
CH	--	8.3	11.6	10.8	2.23
AFt	--	6.5	11.2	10.8	1.78
AFm-Mc	--	0.3	0.5	0.6	2.22
ACn (mainly C-S-H gel)	--	47.0	47.0	47.0	--
FW	31.0	14.3	14.3	14.3	1.00

^a^ This binder also contains at t_0_: 0.6 wt% of gypsum and 0.3 wt% of quartz. ^b^ These capillaries also contain 0.5 wt% of quartz.

**Table 2 materials-14-06953-t002:** RQPA results (wt%), including ACn and FW contents, for the w/c = 0.65 samples referred To 100 g of paste.

	t_0_ ^a^	t = 50 d ^b^ ϕ = 0.50 mm	t = 50 d ^b^ ϕ = 1.00 mm	t = 50 d Cylinder
C_3_S	38.8	3.6	2.4	3.1
C_2_S	7.2	5.2	4.5	4.3
C_4_AF	7.3	1.6	1.3	1.4
C_3_A	3.2	0.9	0.6	0.8
$C\bar{S}$H_0.5_	0.8	--	--	--
$C\bar{C}$	2.2	2.0	1.9	5.5
CH	--	11.1	12.4	10.0
AFt	--	10.2	11.1	9.8
AFm-Mc	--	0.6	1.1	0.8
ACn (mainly C-S-H gel)	--	39.3	39.3	39.3
FW	39.4	25.0	25.0	25.0

^a^ This binder also contains at t_0_: 0.5 wt% of gypsum and 0.3 wt% of quartz. ^b^ These capillaries also contain 0.5 wt% of quartz.

**Table 3 materials-14-06953-t003:** Segmentation results for the three sets of components (vol%) in the four analysed tomograms. The grey-scale ranges used for global segmentation are given in italics.

	Pore (vol%)	HP (vol%)	UHP (vol%)	rHP (vol%)	rUHP (vol%)
ϕ = 0.5 mm, w/c = 0.45	1.5	89.1	9.5	90.4	9.6
*Thresholds*	*0–61*	*62–124*	*125–255*		
ϕ = 0.5 mm, w/c = 0.65	8.3	85.8	5.6	93.9	6.1
*Thresholds*	*0–59*	*60–132*	*133–255*		
ϕ = 1.0 mm, w/c = 0.45	1.5	89.6	8.9	91.0	9.0
*Thresholds*	*0–64*	*65–102*	*103–255*		
ϕ = 1.0 mm, w/c = 0.65	2.4	90.2	7.4	92.4	7.6
*Thresholds*	*0–54*	*55–98*	*99–255*		

**Table 4 materials-14-06953-t004:** Comparison of the RQPA analysis results, grouped in the two sets of components, with those derived from μCT.

	RQPA		μCT	
	HP (vol%)	UHP (vol%)	rHP (vol%)	rUHP (vol%)
ϕ = 0.5 mm, w/c = 0.45	89.1	10.9	90.4	9.6
ϕ = 1.0 mm, w/c = 0.45	92.7	7.3	91.0	9.0
ϕ = 0.5 mm, w/c = 0.65	94.2	5.8	93.9	6.1
ϕ = 1.0 mm, w/c = 0.65	95.5	4.5	92.4	7.6

## Data Availability

All μCT, LXRPD and TA raw data analysed in this article are openly deposited in Zenodo at 10.5281/zenodo.5599198.

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
