# Peer review of "Accuracy in Cement Hydration Investigations: Combined X-ray Microtomography and Powder Diffraction Analyses"

_materials, 2021, doi:10.3390/ma14226953_

Round 1
Reviewer 1 Report
Comments on materials-1461264
The paper deals with the Cement Hydration by combing X-ray Microtomography and Powder Diffraction Analyses. How to determine the cement hydration degree is the fundamental study in cement research. This study is interesting. I think it can be accepted after a moderate revision.
1) In the title, only one method X-ray Rietveld to determine the cement hydration. However, the authors themselves think “However, accurate measurements of some microstructural features, including porosity and amorphous content developments, is more challenging.”. This is definitely true. The cement hydration is very complex since many kinds of crystalline or amorphous phases are intermixed together. Therefore, one can never measure the cement hydration process just by one single method. In this regard, many testing methods are usually used together to measure the cement hydration, such as XRD, NMR, TG/DTA/SEM/MIP/SEM, etc, in order to prove the accuracy of the testing results. However, why did the author just use one single method to determine the cement hydration process? X-ray Microtomography is a method related with structure detection rather than a method to measure the hydration products directly.
2) Besides, since the authors mentioned C-S-H again and again. The authors should describe the advantage as well the disadvantage of X-ray Rietveld, can this method detect the amorphous phases such as C-S-H accurately?
3) In introduction, the first sentence “Anhydrous cements are multi-phase materials and they can be well understood when they are studied by a multi-technique approach [1], mainly” include three aspects, those are, XRD, XRF, and textural characterization…” is not so accurate and comprehensive. The authors should introduce other popular methods to determine the hydration degree, such as NMR and TG techniques. Since NMR is so popular a method in recent decades, the authors should compare their hydration results to those by NMR. I think the recent studies Comparison between the effects of phosphorous slag and fly ash on the C-S-H structure, long-term hydration heat and volume deformation of cement-based materials and Investigation of microstructure of C-S-H and micro-mechanics of cement pastes under NH4NO3 dissolution by 29Si MAS NMR may help to improve will strengthen this paper very well. I think the authors should at least introduce or mention other important testing methods as well, rather than just mentioning X-ray Rietveld, only the introduction be revised in this way can give the potential readers a comprehensive impression how to test the cement hydration.
4) Also, the authors should compare their TG results with those from others. In addition, the authors should explain why they adopted the three methods rather than NMR and others.
The authors used TG too, but the TG results in this study is a little vague. May I ask what is the purpose of using TG? I think TG is also an effective tool to evaluate the cement hydration. Can the authors compare their TG results with others? Such as Effect of cement type and limestone
particle size on the durability of steam cured self-consolidating concrete,
5) The authors attributed the loss from 45° to 550°C to free water , however, the necessary evidence or references should be proved to prove this claim.
6) The basis properties of raw materials should be given, a simple “Full characterization of this cement has been reported elsewhere [24].” is far not enough.
7) The conclusion part should include the main findings of this study, rather than a simple introduction of their study in one or two sentences.
Author Response
We thank him/her for the time to improve the quality of the manuscript. The answers are given next and all changes will be highlighted in blue in the revised version to be easily followed if needed.

Reviewer 2 Report
Dear Authour,
Kindly see the following comments,
1) Improve overall English in the paper
2) Check all references cited in the text and at the end as per the format of the journal.
3) See all figures and tables are up to the mark and cited in the text well.
Author Response
We thank him/her for the effort to improve this manuscript. All modifications in blue in the revised version.
- The English has been thoroughly edited.
2. All reference have been checked and they cited in the text. However, some minor errors in the reference format, not conforming the guidelines of the journal, have been corrected.
3. We have double-checked the figures and tables and they are all well cited and with the required quality.
Reviewer 3 Report
This work presents a combined X-ray microtomography and powder diffraction analyses on cement hydration, which is a complex set of processes. It is shown that the water capillary porosity is segmented within the hydrated component fractions, which provide a method to accurately determine selected hydration features like the hydration degree development of the amorphous fraction of supplementary cementitious materials. The experiments are well designed and conducted. The results are clearly presented and the data are open access, which is a good thing for readers to follow their researches. I have no concerns to be addressed for this paper, I think the paper is suitable to be published on Materials.
Author Response
We thank him/her for his time and feedback. No corrections are needed.
Reviewer 4 Report
1) What does the symbol "Q" mean in diagram 2c and 3c?
2)Why the pore vol.% for a sample of 0.5 mm and w / c = 0.65 is as high as 8.c? It is practically four times more than for other samples? (Table 3)
3) Why the authors chose the maturing time of the cement paste for 50 days? After all, they used normal cement 42.5, where the manufacturer states the maturation time for 28 days?
4) The areas covered with pores are not visible in the internal sections (Fig. 7)? Why? Were the samples not thoroughly mixed and formed? Only free water and / or surface porosity can be observed?
Author Response
We thank him/her for the time to improve the quality of the manuscript. The answers are given in the attached file, and all changes are highlighted in blue in the revised version to be easily followed.

Round 2
Reviewer 1 Report
This revised version cannot well address my previous comments, since many of my questions are still not well answered. So, I suggest a continuous revision and ask the authors if they could give a response to comments one item by one item.
I think the studies I suggested last time needs to be read carefully, because 1 H NMR is only used to analyze pore structure and cannot directly be adopted to analyze hydration products. Therefore, the author's revision this time is not accurate.
I will propose acceptance if my comments are fully and carefully addressed. I attached my previous comments here.
Comments on materials-1461264
The paper deals with the Cement Hydration by combing X-ray Microtomography and Powder Diffraction Analyses. How to determine the cement hydration degree is the fundamental study in cement research. This study is interesting. I think it can be accepted after a moderate revision.
1) In the title, only one method X-ray Rietveld to determine the cement hydration. However, the authors themselves think “However, accurate measurements of some microstructural features, including porosity and amorphous content developments, is more challenging.”. This is definitely true. The cement hydration is very complex since many kinds of crystalline or amorphous phases are intermixed together. Therefore, one can never measure the cement hydration process just by one single method. In this regard, many testing methods are usually used together to measure the cement hydration, such as XRD, NMR, TG/DTA/SEM/MIP/SEM, etc, in order to prove the accuracy of the testing results. However, why did the author just use one single method to determine the cement hydration process? X-ray Microtomography is a method related with structure detection rather than a method to measure the hydration products directly.
2) Besides, since the authors mentioned C-S-H again and again. The authors should describe the advantage as well the disadvantage of X-ray Rietveld, can this method detect the amorphous phases such as C-S-H accurately?
3) In introduction, the first sentence “Anhydrous cements are multi-phase materials and they can be well understood when they are studied by a multi-technique approach [1], mainly” include three aspects, those are, XRD, XRF, and textural characterization…” is not so accurate and comprehensive. The authors should introduce other popular methods to determine the hydration degree, such as NMR and TG techniques. Since NMR is so popular a method in recent decades, the authors should compare their hydration results to those by NMR. I think the recent studies Comparison between the effects of phosphorous slag and fly ash on the C-S-H structure, long-term hydration heat and volume deformation of cement-based materials and Investigation of microstructure of C-S-H and micro-mechanics of cement pastes under NH4NO3 dissolution by 29Si MAS NMR may help to improve will strengthen this paper very well. I think the authors should at least introduce or mention other important testing methods as well, rather than just mentioning X-ray Rietveld, only the introduction be revised in this way can give the potential readers a comprehensive impression how to test the cement hydration.
4) Also, the authors should compare their TG results with those from others. In addition, the authors should explain why they adopted the three methods rather than NMR and others.
The authors used TG too, but the TG results in this study is a little vague. May I ask what is the purpose of using TG? I think TG is also an effective tool to evaluate the cement hydration. Can the authors compare their TG results with others? Such as Effect of cement type and limestone
particle size on the durability of steam cured self-consolidating concrete,
5) The authors attributed the loss from 45° to 550°C to free water , however, the necessary evidence or references should be proved to prove this claim.
6) The basis properties of raw materials should be given, a simple “Full characterization of this cement has been reported elsewhere [24].” is far not enough.
7) The conclusion part should include the main findings of this study, rather than a simple introduction of their study in one or two sentences.
Author Response
Thanks. The answers are detailed in the attached letter.

Round 3
Reviewer 1 Report
The authors did a good revision work, all my previous comments have been addressed one by one. I think it can be accepted now.